**Research**

# Lifetime socioeconomic circumstances and chronic pain in later adulthood: findings from a British birth cohort study

Matthew A Jay, [1,2] Rebecca Bendayan, [3,4] Rachel Cooper, [4] Stella G Muthuri [4]

¹GOS Institute of Child Health, University College London, London, UK
²Department of Anaesthesia and Pain Medicine, Great Ormond Street Hospital For Children NHS Foundation Trust, London, UK
³Department of Biostatistics and Health Informatics, King's College London, Institute of Psychiatry, Psychology and Neuroscience, London, UK
⁴MRC Unit for Lifelong Health and Ageing, University College London, London, UK

**Correspondence to**
Mr Matthew A Jay;
matthew.jay.15@ucl.ac.uk

## ABSTRACT

**Objectives** To investigate associations between a range of different indicators of socioeconomic position (SEP: occupational class, education, household overcrowding and tenure, and experience of financial hardship) across life and chronic widespread and regional pain (CWP and CRP) at age 68.

**Design** Prospective birth cohort; the Medical Research Council National Survey of Health and Development.

**Setting** England, Scotland and Wales.

**Participants** Up to 2378 men and women who have been followed-up since birth in 1946 to age 68.

**Primary outcome measures** On the basis of their self-report of pain at age 68, participants were classified as: CWP (American College of Rheumatology criteria), CRP (pain of at least 3 months' duration but that does not meet the definition of CWP), other pain (<3 months in duration) or no pain.

**Results** At age 68, the prevalence of CWP was 13.3% and 7.8% in women and men, respectively, and that of CRP was 32.3% and 28.7% in women and men, respectively. There was no clear evidence that indicators of SEP in childhood or later adulthood were associated with pain. Having experienced (vs not) financial hardship and being a tenant (vs owner-occupier) in earlier adulthood were both associated with an increased risk of CWP; for example, moderate hardship adjusted relative risk ratio (RRR_adj) 2.32 (95% CI: 1.19 to 4.52) and most hardship RRR_adj 4.44 (95% CI: 2.02 to 9.77). Accumulation of financial hardship across earlier and later adulthood was also associated with an increased risk of CWP.

**Conclusions** Consideration of socioeconomic factors in earlier adulthood may be important when identifying targets for intervention to prevent CWP in later life.

## Strengths and limitations of this study

► We used data from a sample of older adults in Great Britain who were representative of the general population born at a similar time.
► Studying a wide range of indicators of SEP over a 60-year-period provided a unique insight into the associations between SEP and CWP and CRP at age 68.
► Prospective data collection across life reduced the risk of misclassification bias.
► The study might have been subject to survivor bias with the result that associations may be biased towards the null.

or longer that does not otherwise meet these ACR criteria. In a systematic review, the prevalence of chronic pain among adults in the UK was estimated to be 44% (95% CI: 38% to 49%) and that of CWP to be 14% (95% CI: 12% to 16%),[4] an estimate that is similar in other countries.[5 6] Understanding risk factors for CWP and CRP, such as socioeconomic position (SEP) on which there is currently limited population-level evidence,[7] is, therefore, important to develop preventive public health strategies. Pain is a biopsychosocial phenomenon, meaning that it is known to be affected by a range of biological, psychological and social factors.[8] SEP is, therefore, an important target for study and potential intervention.

Few studies have examined associations between SEP and CWP, CRP or a combined measure of chronic pain; they have largely been cross-sectional, tending to treat SEP as a confounder.[9–14] They mostly examine only one or two indicators of SEP despite its multifaceted nature[15–17] with different aspects of SEP potentially related to CWP and CRP in different ways. Few also consider the role of SEP across life.[18–20] Of those that do, Macfarlane *et al*[18] and Power *et al*,[19] using the 1958

## INTRODUCTION

Chronic widespread and regional pain (CWP and CRP) are leading causes of disability globally,[1] particularly in later adulthood.[2] CWP is defined according to the American College of Rheumatology (ACR) criteria[3] as pain present for 3 months or longer, both above and below the waist; on both the left and right side of the body and in the axial skeleton. CRP is any pain present for 3 months

British birth cohort, found associations between lower SEP (paternal and own occupational class during childhood and earlier adulthood, respectively) and higher risk of CWP at age 44–45. Macfarlane *et al*[18] also found that lower occupational class was associated with an increased risk of CRP at specific sites; many of these associations, however, were attenuated after adjustment. Goosby,[20] using the US Health and Retirement Study, found that higher education and income were associated with lower risk of chronic pain (ie, arthritis/rheumatism; back or neck problems or received treatment in the past 12 months; frequent or severe headaches and any other chronic pain) in people aged 18–64. The extent to which SEP across life is associated with CWP or CRP in participants beyond midlife remains to be determined. Taking a life-course approach, by investigating SEP at different life stages, could help to identify new opportunities for intervention as the accumulation of exposure to social stressors across life may be associated with an increased risk of CWP and CRP at older ages.[21]

There are important gaps in knowledge. First, the multi-faceted nature of SEP has been neglected in research into the association between SEP and CWP and CRP. Our first aim was, therefore, to investigate whether a range of different SEP indicators across childhood and adulthood were associated with CWP, CRP or other pain at age 68. Examining CWP and CRP is important as they may have different underlying aetiologies. Second, associations between SEP and CWP or CRP in later adulthood cannot be fully understood without considering how SEP operates across life. We, therefore, also aimed to test whether observed associations are cumulative. We hypothesised that (1) lower levels of each SEP indicator would be associated with an increased risk of CWP (but not necessarily CRP on the basis of the weaker associations previously observed[18]) in later adulthood and (2) accumulation of low SEP across life would also be associated with increased risk of CWP.

## METHODS

### Sample

Data were drawn from the Medical Research Council (MRC) National Survey of Health and Development (NSHD), a nationally representative[22 23] longitudinal study of a sample (n=5362) of single births during the first week of March 1946 in Great Britain.[24] Data collection occurred approximately every 2 years during childhood and every 5–10 years during adulthood, with 24 data collection rounds up to 2014–2015.[22 24 25] Details on overall participation rates, which were generally high, are available elsewhere.[22 26] Of the 2942 people in the targeted sample for the most recent data collection at age 68, in 2014–2015, 2453 (83.4%) completed a postal questionnaire. No attempt was made to contact the remaining 2420 study members: 957 (17.8%) had already died, 620 (11.6%) had previously withdrawn from the study, 448 (8.3%) had emigrated and were no longer in contact with

the study and 395 (7.4%) had been untraceable for more than 5 years. Participants who responded were eligible for the present study if they had provided responses to questions on pain (n=2378).

### Patient and public involvement statement

Participants have a lifelong association with NSHD. Over the 70 years of the study, the research team has increasingly involved participants, in line with changing norms about conducting cohort studies, starting at age 16 (in 1962) with the annual dissemination of study findings in birthday cards. Participants receive personal letters whenever they raise queries or provide additional comments, including suggestions for new topics to study. In the last 10 years, the research team has increased the level of participant involvement through invitations to study events (65th and 70th birthday celebrations), public engagement activities and focus groups to discuss clinical sub-studies. When piloting new questionnaires and assessments, we recruit patients from general practices or the University College London Hospital patient public involvement group and take into account their feedback when designing the main-stage fieldwork.

### Ascertainment of pain at age 68

Participants were asked, 'In the *last month*, have you had any ache or pain which has lasted for *one day or longer*? (Please do not include pain occurring only during the course of a feverish illness, such as flu [*sic*])' (emphasis in original). Those who answered yes were then asked whether they had been aware of this pain for at least 3 months and asked to shade on a four-view body manikin all the areas where they felt pain. Responses to these questions were used to classify participants as having CWP (defined according to the ACR criteria[3]). Those who reported pain of at least 3 months' duration but did not meet the CWP definition were classified as CRP and those reporting pain of less than 3 months duration were categorised as having other pain.[27]

### Indicators of SEP

A range of SEP indicators were selected for study *a priori* in order to capture different facets of the SEP construct including psychosocial as well as material aspects.[16 17] These indicators were assessed prospectively across life and for the purposes of these analyses were grouped into three life stages (childhood [age 0 to 15]; earlier adulthood [age 26 to 53] and later adulthood [age 60 to 64]). Indicators of childhood SEP were father's occupational class, paternal and maternal education, household overcrowding, housing tenure and lack of household amenities. Where possible, measures were taken at age 4, except for lack of amenities, which was ascertained at age 2. However, if data were missing at age 4, comparable measures at ages 11 or 15 were used. See table 1 for details of categorisation of each measure.

Indicators of SEP in earlier adulthood (ie, 26 to 53) were highest educational level attained by age 26,

**Table 1** Indicators of socioeconomic position used in analyses.

| Indicator | Definition | Age ascertained (ages used if missing at main age) |
|---|---|---|
| **Childhood** | | |
| Paternal occupational class | Registrar General's Social Classification: I and II—professional and managerial, and technical IIINM—skilled non-manual IIIM—skilled manual IV and V—partly-skilled and unskilled | 4 (or 11 [n=48] or 15 [n=17]) Total=2258 |
| Paternal and maternal education | Higher Secondary only Primary/less | 6 (paternal=2117 and maternal=2099) |
| Household overcrowding | >2 persons per room in the household | 4 (or 11 [n=34] or 15 [n=13]) Total=2298 |
| Housing tenure | Owner-occupier versus tenant (includes private and public tenants) | 4 (or 11 [n=42] or 15 [n=12]) Total=2300 |
| Lack of household amenities | Lacks access to hot running water, own bathroom or own kitchen (score indicating lacks 0 to all 3). | 2 (or 11 [n=157] or 15 [n=11]) Total=2326 |
| **Earlier adulthood** | | |
| Own occupational class | Registrar General's Social Classification (categorised as above) | 53 (or 43 [n=123] or 36 [n=59]) Total=2219 |
| Home ownership | Owner-occupier versus tenant (includes private and public tenants) | 53 (or 43 [n=142] or 36 [n=29]) Total=2294 |
| Highest educational level attained | Degree or higher A level or equivalent or less | 26 (n=2297) |
| Financial hardship in the previous year | None (no hardship) Minimal (hard to manage but does not report going without necessities or unable to pay bills) Moderate (hard to manage and either went without necessities or unable to pay bills) Most (hard to manage, went without necessities and unable to pay bills) | 43 (n=2150) |
| **Later adulthood** | | |
| Home ownership | As in earlier adulthood | 60–64 (n=2038) |
| Financial hardship | As in earlier adulthood | 60–64 (n=1892) |

financial hardship at age 43 and own occupational class and home ownership at age 53 (or at 43 or 36 if missing at 53) (see table 1). Indicators of SEP in later adulthood (age 60–64) were home ownership and financial hardship (see table 1).

## Covariates

Potential confounders were identified from the existing literature.[10 18 28–31] Body mass index (BMI) (kg m$^{-2}$) was calculated using measures of height and weight recorded by trained nurses at age 60–64. Cigarette smoking status (current, ex or never), participation in sports or vigorous leisure activities in the previous 4 weeks (inactive, 1–4 times per month, or five or more times per month) and marital status (never married, married/civil partnership, separated/divorced and widowed) were all self-reported at age 68. Participants were asked how often they consumed alcoholic drinks in the last year and to record

the number of alcoholic drinks (beers, wines and spirits) they drank in the last 7 days. Units were summed to create an overall measure of alcohol consumptions in units in the previous week at age 68. Affective symptoms were measured using General Health Questionnaire-28 (GHQ-28)[32] score at age 60–64. Items were scored 0–0–1–1 and then summed, and caseness was identified based on a cut-off of five or more.[32]

## Statistical methods

To examine the associations of each SEP indicator with CWP, CRP and other pain at age 68, a series of multinomial logistic regression models were specified. For each SEP indicator, a crude model was first constructed and adjusted for sex. An interaction term between each SEP indicator and sex was entered and subsequent models were sex-stratified if this term was statistically significant (likelihood ratio test, $p<0.05$). Categories of covariates

were entered as follows: First, BMI and health behaviours (smoking, physical activity and alcohol consumption) were entered. Second, the adjustment was made for affective symptoms (GHQ-28 caseness). Finally, a fully adjusted model that included all covariates was estimated.

To ascertain whether the accumulation of low levels of SEP was associated with CWP, CRP or other pain, we derived ordinal indices that indicated, at each time point, whether the participant was in lower or higher SEP (these scores assume equal weight should be given to SEP at each life stage). We only generated these indices for indicators of SEP that had been found at any life stage to be associated with CWP, CRP or other pain and had been measured at multiple life stages. These indices were entered into regression models and adjustments were carried out in the same order as above.

Models were fit with maximum likelihood and results are presented as relative risk ratios (RRRs) with 95% CIs. Overall statistical significance of each SEP indicator in each model was assessed using the likelihood ratio test. Where covariates were missing (GHQ-28 score [n=494, 20.8%]; BMI [n=472, 19.8%]; alcohol consumption [n=243, 10.2%]; marital status [n=45, 1.9%] and smoking status [n=22, 0.9%]), these were imputed using multiple imputations with chained equations[33] with 40 imputed datasets. The outcome and exposure variables were included in the imputation model but not themselves imputed.[34] Parameter estimates were combined using Rubin's rules. Sensitivity analyses exploring these models with only complete cases were performed.

All data management and analysis was conducted in R[35] with RStudio[36] and the packages *mice*[33] and *nnet*.[37]

## RESULTS
The characteristics of the 2378 participants included in analyses are presented in table 2. The prevalence of CWP among women and men was 13.3% and 7.8%, respectively. In women, the prevalence of CRP was 32.3% and other pain was 12.5%. In men, these figures were 28.7% and 15.2%, respectively. The number and proportion of individuals in the lowest SEP categories are given in table 2, with a full breakdown by sex in online supplementary tables 1–3.

There was no evidence of associations between any of the indicators of childhood SEP and CWP, CRP or other pain at age 68 (table 3; models at all stages of adjustment in supplementary table 4).

Table 4 (online supplementary table 5) shows the RRRs of CWP, CRP and other pain (vs no pain) obtained from regression models for each indicator of SEP in earlier adulthood. There were sex differences in the associations between educational level and pain (education–sex interaction, p<0.01). Women with lower levels of education were more likely to report CRP at age 68 compared with those with a degree or higher (RRR 1.93 [95% CI: 1.01 to 3.69]) and less likely to report other pain (RRR 0.54 [0.29 to 1.00]). Men with lower educational levels

were more likely to report CWP than men with a degree or higher (2.68 [95% CI: 1.12 to 6.45]). Financial hardship in earlier adulthood was associated with higher risk of CWP at age 68; fully-adjusted RRR of CWP was 1.21 (95% CI: 0.67 to 2.17), 2.32 (95% CI: 1.19 to 4.52) and 4.44 (95% CI: 2.02 to 9.77) for minimal, moderate and most financial hardship, respectively, compared with no financial hardship. Similar trends were observed for CRP and other pain but the RRRs were smaller and not statistically significant. In a sex-adjusted model, there was some evidence of an association between the lowest occupational classes and an increased risk of CWP (RRR 1.52 [95% CI: 0.99 to 2.34]), which was attenuated in the fully-adjusted model (RRR 1.22 [95% CI: 0.77 to 1.92]). Finally, being a tenant (vs being an owner-occupier) was associated with an increased risk of CWP at age 68. In a sex-adjusted model, the RRR was 1.81 (95% CI: 1.21 to 2.72), which was slightly attenuated to 1.62 (1.06 to 2.49) after adjustment for all covariates.

Neither indicator of SEP in later adulthood (home ownership and financial hardship) was associated with pain at age 68 (table 4 and supplementary table 6).

As tenant status and financial hardship were the only two variables measured at multiple time points that were associated with CWP or CRP, these two were taken forward into the accumulation analyses. The number and proportion of participants in each group at the earliest and latest time points are cross-tabulated in supplementary table 7. Online supplementary table 8 shows the number and proportion of participants at each level of the derived accumulation indices.

Results from multinomial logistic regression models examining the cumulative associations between tenant status, financial hardship and CWP, CRP and other pain are shown in table 5 (online supplementary table 9). These show that accumulation of tenant status over life was not associated with pain at age 68. By contrast, accumulation of financial hardship across adulthood was associated with an increased risk of CWP (hardship at one point reported vs no hardship RRR 1.93 [95% CI: 1.11 to 3.35]; hardship at both points RRR 3.90 [95% CI: 1.20 to 12.64]).

Sensitivity analyses revealed no major differences to the models fit using imputed data (directions of associations were the same and magnitudes were similar) and are not, therefore, presented.

## DISCUSSION
We aimed to investigate whether different SEP indicators at different life stages were associated with risk of reporting CWP, CRP or other pain at age 68. Experience of financial hardship and tenant status in earlier adulthood were both associated with an increased risk of CWP at age 68. Lower educational attainment was also associated with an increased risk of CWP and CRP in men and women, respectively. There was no evidence of associations between any SEP indicator in childhood

**Table 2** Characteristics of participants from the MRC National Survey of Health and Development included in analyses (sample restricted to those with data on pain at age 68 and at least one indicator of SEP [maximum n=2378]).

| | | Women (n=1140) | Men (n=1238) |
|---|---|---|---|
| **Pain at age 68** | | | |
| CWP | | 164 (13.3%) | 89 (7.8%) |
| CRP | | 400 (32.3%) | 327 (28.7%) |
| Other pain | | 155 (12.5%) | 173 (15.2%) |
| No pain | | 519 (41.9%) | 551 (48.3%) |
| **BMI at age 60–64, n=1906** | | | |
| Mean (SD) (kg m$^{-2}$) | | 27.8 (5.2) | 27.8 (4.0) |
| **Smoking at age 68, n=2356** | | | |
| Current | | 96 (7.8%) | 112 (9.9%) |
| Ex-smoker | | 653 (53.2%) | 680 (60.2%) |
| Never | | 478 (39.0%) | 337 (29.9%) |
| Alcohol units (Median [IQR]) at age 68, n=2135 | | 2.0 (0.0 to 6.0) | 7.0 (2.0 to 15.0) |
| **Leisure time physical activity at age 68, n=2352** | | | |
| Inactive | | 733 (59.7%) | 682 (60.7%) |
| One to four times/month | | 163 (13.3%) | 120 (10.6%) |
| Five or more times/month | | 330 (26.9%) | 324 (28.7%) |
| **Symptoms of anxiety and depression (GHQ-28) at age 60–64, n=1884** | | | |
| Yes (≤4) | | 222 (22.4%) | 115 (12.9%) |
| No (>4) | | 768 (77.6%) | 779 (87.1%) |
| **Marital status at age 68, n=2333** | | | |
| Single | | 39 (3.2%) | 47 (4.2%) |
| Married | | 856 (70.7%) | 912 (81.2%) |
| Separated/Divorced | | 167 (13.8%) | 109 (9.7%) |
| Widowed | | 148 (12.2%) | 55 (4.9%) |
| **Childhood SEP (age 0–15)** | | | |
| Father's occupational class | Class IV–V | 291 (24.8%) | 260 (24.0%) |
| Home ownership | Tenant status | 826 (69.0%) | 773 (70.1%) |
| Lack of amenities | Lacks three amenities | 48 (4.0%) | 47 (4.2%) |
| Overcrowding | Overcrowded | 67 (5.6%) | 39 (3.5%) |
| Maternal education | Primary or less | 817 (74.4%) | 742 (72.8%) |
| Paternal education | Primary or less | 740 (67.8%) | 667 (65.9%) |
| **Earlier adulthood SEP (age 26–53)** | | | |
| Educational level | A level or equivalent or less | 1116 (93.5%) | 913 (82.8%) |
| Financial hardship | Most hardship | 31 (2.8%) | 23 (2.3%) |
| Occupational class | Class IV–V | 193 (17.0%) | 93 (8.6%) |
| Home ownership | Tenant status | 128 (10.7%) | 116 (10.6%) |
| **Later adulthood SEP (age 60/64)** | | | |
| Home ownership | Tenant status | 283 (26.1%) | 307 (32.2%) |
| Financial hardship | Most hardship | 11 (1.1%) | 19 (2.1%) |

For brevity, descriptive statistics for only the most adverse category of each SEP indicator are presented; a complete breakdown of the distribution of each SEP indicator by sex is given in online supplementary tables 1 and 2. Numbers on each covariate vary due to missing data.

BMI, body mass index; CRP, chronic regional pain; CWP, chronic widespread pain; GHQ-28, General Health Questionnaire-28-item; SEP, socioeconomic position.

**Table 3** Associations between indicators of socioeconomic position in childhood and pain at age 68.

| Exposure | n | Model | Exposure level | CWP versus no pain RRR (95% CI) | CRP versus no pain RRR (95% CI) | Other pain versus no pain RRR (95% CI) | P value |
|---|---|---|---|---|---|---|---|
| Father's occupational class (ref=I–II) | 2258 | 1 | IIINM | 1.11 (0.73 to 1.69) | 0.94 (0.71 to 1.26) | 0.90 (0.62 to 1.31) | 0.87 |
| | | | IIIM | 1.06 (0.71 to 1.57) | 1.09 (0.84 to 1.41) | 0.87 (0.62 to 1.24) | |
| | | | IV–V | 1.29 (0.87 to 1.92) | 1.00 (0.76 to 1.32) | 1.00 (0.70 to 1.42) | |
| | | 2 | IIINM | 1.05 (0.68 to 1.62) | 0.93 (0.70 to 1.24) | 0.90 (0.62 to 1.32) | 0.97 |
| | | | IIIM | 0.86 (0.57 to 1.29) | 1.01 (0.77 to 1.31) | 0.83 (0.58 to 1.19) | |
| | | | IV–V | 1.05 (0.69 to 1.59) | 0.93 (0.70 to 1.23) | 0.98 (0.68 to 1.41) | |
| Home ownership (ref=owner occupier) | 2300 | 1 | Tenant | 1.06 (0.78 to 1.43) | 1.05 (0.85 to 1.29) | 1.02 (0.77 to 1.34) | 0.96 |
| | | 2 | Tenant | 0.88 (0.64 to 1.21) | 0.97 (0.78 to 1.20) | 0.98 (0.74 to 1.30) | 0.88 |
| Lack of amenities (ref=lacks 0) | 2326 | 1 | Lacks one or more | 1.21 (0.92 to 1.60) | 1.03 (0.85 to 1.24) | 0.99 (0.77 to 1.27) | 0.57 |
| | | 2 | Lacks one or more | 1.14 (0.85 to 1.52) | 1.01 (0.83 to 1.22) | 0.98 (0.76 to 1.26) | 0.81 |
| Overcrowding (ref=not overcrowded) | 2298 | 1 | Overcrowded | 1.12 (0.59 to 2.11) | 1.17 (0.75 to 1.82) | 0.72 (0.36 to 1.44) | 0.56 |
| | | 2 | Overcrowded | 1.01 (0.52 to 1.95) | 1.13 (0.72 to 1.77) | 0.71 (0.35 to 1.44) | 0.64 |
| Maternal education (ref=diploma/degree) | 2117 | 1 | Secondary | 1.06 (0.58 to 1.95) | 0.76 (0.51 to 1.13) | 0.56 (0.34 to 0.95) | 0.40 |
| | | | Primary | 1.08 (0.65 to 1.82) | 0.82 (0.59 to 1.13) | 0.75 (0.49 to 1.13) | |
| | | 2 | Secondary | 1.01 (0.54 to 1.88) | 0.74 (0.50 to 1.11) | 0.57 (0.34 to 0.97) | 0.38 |
| | | | Primary | 0.94 (0.55 to 1.61) | 0.76 (0.54 to 1.06) | 0.73 (0.48 to 1.11) | |
| Paternal education (ref=diploma/degree) | 2099 | 1 | Secondary | 1.12 (0.66 to 1.87) | 1.26 (0.89 to 1.78) | 0.83 (0.53 to 1.30) | 0.68 |
| | | | Primary | 1.11 (0.73 to 1.67) | 1.07 (0.81 to 1.42) | 0.84 (0.60 to 1.20) | |
| | | 2 | Secondary | 0.99 (0.58 to 1.69) | 1.19 (0.84 to 1.70) | 0.80 (0.51 to 1.27) | 0.64 |
| | | | Primary | 0.90 (0.58 to 1.39) | 0.97 (0.72 to 1.30) | 0.81 (0.57 to 1.16) | |

Model 1: sex-adjusted and Model 2: adjusted for sex, BMI, smoking, alcohol, leisure-time physical activity, GHQ-28 caseness and marital status.
CRP, chronic regional pain; CWP, chronic widespread pain.

or later adulthood and any pain outcome at age 68. Our second aim was to assess whether observed associations were cumulative across different life stages and we found that there was a cumulative association between financial hardship during earlier and later adulthood and increased risk of CWP at age 68.

Our finding of stronger and more consistent associations between indicators of SEP in earlier adulthood and pain than with indicators of SEP in childhood or later adulthood, suggests that earlier adulthood may be a sensitive period. Financial hardship appears to be particularly important. This variable captures both subjective and objective elements of hardship (ie, feeling unable to manage or being unable to pay bills, or both). There may, therefore, be a combination of psychosocial and material elements that are associated with an increased risk of CWP. It is also especially interesting that financial hardship in later adulthood was not independently associated with CWP but when combined with the same variable in earlier adulthood, a clear inverse association was observed. This suggests that, despite the lack of strong evidence of an association between financial hardship in later adulthood and CWP at 68, intervening to reduce the burden of hardship in later as well as earlier adulthood

may prove beneficial in reducing the prevalence of CWP in older age.

The interaction between education and sex also requires explanation. An increased risk of CWP or CRP with lower levels of education is in line with our initial hypotheses but it is not clear why lower education was associated with an increased risk of CWP among men but with an increased risk of CRP among women. The results may be partly explained by the fact that the binary categorisation of education used in our analyses captured more of the relevant variability in educational attainment among men than women; in our sample, 6.5% of women had a degree or higher compared with 17.2% of men (see online supplementary table 2). In addition, it may be the case that among this generation, comparing those who have a degree or higher with those with a lower level of educational attainment provides a more appropriate level of differentiation between higher and lower SEP among men than women. Previous work using the NSHD has shown that the normative work-family life course patterns for women was one of forming families earlier in life when compared with men.[38 39] Therefore, among this generation, achieving a university degree may have been less likely to have provided women than men with access

**Table 4** Associations between indicators of socioeconomic position in earlier and later adulthood and pain at age 68.

| Exposure | n | Model | Exposure level | CWP versus no pain RRR (95% CI) | CRP versus no pain RRR (95% CI) | Other pain versus no pain RRR (95% CI) | P value |
|---|---|---|---|---|---|---|---|
| **Earlier adulthood** | | | | | | | |
| Educational level (women)* (ref=degree) | 1194 | 2 | A level or equivalent or less | 0.94 (0.44 to 2.01) | 1.93 (1.01 to 3.69) | 0.54 (0.29 to 1.00) | <0.01 |
| Educational level (men)* (ref=degree) | 1103 | 2 | A level or equivalent or less | 2.68 (1.12 to 6.45) | 1.07 (0.74 to 1.56) | 1.45 (0.88 to 2.40) | 0.06 |
| Financial hardship (ref=none) | 2150 | 1 | Minimal | 1.37 (0.78 to 2.42) | 1.13 (0.75 to 1.70) | 0.98 (0.56 to 1.71) | <0.01 |
| | | | Moderate | 2.61 (1.38 to 4.95) | 1.31 (0.76 to 2.28) | 1.31 (0.64 to 2.67) | |
| | | | Most | 5.38 (2.56 to 11.30) | 1.81 (0.89 to 3.72) | 1.67 (0.67 to 4.18) | |
| | | 2 | Minimal | 1.21 (0.67 to 2.17) | 1.09 (0.72 to 1.65) | 0.95 (0.54 to 1.68) | 0.03 |
| | | | Moderate | 2.32 (1.19 to 4.52) | 1.32 (0.75 to 2.30) | 1.34 (0.65 to 2.74) | |
| | | | Most | 4.44 (2.02 to 9.77) | 1.79 (0.86 to 3.74) | 1.75 (0.69 to 4.47) | |
| Occupational class (ref=I–II) | 2219 | 1 | IIINM | 1.15 (0.79 to 1.66) | 0.95 (0.73 to 1.23) | 1.16 (0.83 to 1.61) | 0.15 |
| | | | IIIM | 1.41 (0.92 to 2.19) | 1.35 (1.02 to 1.79) | 0.97 (0.66 to 1.41) | |
| | | | IV–V | 1.52 (0.99 to 2.34) | 1.24 (0.91 to 1.69) | 0.94 (0.61 to 1.45) | |
| | | 2 | IIINM | 1.08 (0.74 to 1.59) | 0.91 (0.70 to 1.19) | 1.13 (0.81 to 1.58) | 0.38 |
| | | | IIIM | 1.29 (0.82 to 2.03) | 1.31 (0.98 to 1.75) | 0.93 (0.63 to 1.38) | |
| | | | IV–V | 1.22 (0.77 to 1.92) | 1.17 (0.85 to 1.61) | 0.93 (0.59 to 1.44) | |
| Home ownership (ref=owner occupier) | 2294 | 1 | Tenant | 1.81 (1.21 to 2.72) | 1.07 (0.77 to 1.48) | 1.33 (0.89 to 1.98) | 0.03 |
| | | 2 | Tenant | 1.62 (1.06 to 2.49) | 1.06 (0.76 to 1.48) | 1.36 (0.90 to 2.05) | 0.11 |
| **Later adulthood** | | | | | | | |
| Home ownership (ref=owner) | 2038 | 1 | Tenant | 1.25 (0.90 to 1.73) | 1.04 (0.83 to 1.30) | 0.90 (0.67 to 1.22) | 0.44 |
| | | 2 | Tenant | 1.07 (0.76 to 1.51) | 0.98 (0.78 to 1.24) | 0.89 (0.66 to 1.21) | 0.83 |
| Financial hardship (ref=none) | 1892 | 1 | Minimal | 1.61 (0.62 to 4.16) | 0.88 (0.40 to 1.94) | 0.77 (0.26 to 2.30) | 0.52 |
| | | | Moderate | 2.03 (0.86 to 4.80) | 1.73 (0.90 to 3.34) | 1.54 (0.66 to 3.62) | |
| | | | Most | 2.26 (0.77 to 6.64) | 1.53 (0.66 to 3.57) | 0.89 (0.24 to 3.20) | |
| | | 2 | Minimal | 1.39 (0.51 to 3.74) | 0.83 (0.37 to 1.87) | 0.77 (0.25 to 2.34) | 0.89 |
| | | | Moderate | 1.47 (0.60 to 3.63) | 1.56 (0.79 to 3.05) | 1.49 (0.63 to 3.54) | |
| | | | Most | 1.71 (0.55 to 5.29) | 1.39 (0.58 to 3.32) | 0.91 (0.25 to 3.34) | |

Model 1: sex-adjusted and Model 2: adjusted for sex, BMI, smoking, alcohol, leisure-time physical activity, GHQ-28 caseness and marital status.
*The models for education are stratified by sex due to a significant interaction term between sex and education (likelihood ratio test, p<0.001).
CRP, chronic regional pain; CWP, chronic widespread pain.

**Table 5** Associations between cumulative exposure to tenant status and financial hardship and pain at age 68.

| Exposure | n | Model | Accumulation | CWP versus no pain RRR (95% CI) | CRP versus no pain RRR (95% CI) | Other pain versus no pain RRR (95% CI) | P value |
|---|---|---|---|---|---|---|---|
| Housing ownership accumulation (ref=always owner) | 1956 | 1 | Tenant at one point | 0.90 (0.61 to 1.33) | 0.91 (0.70 to 1.18) | 0.89 (0.63 to 1.25) | 0.43 |
| | | | Tenant at two points | 1.10 (0.68 to 1.79) | 1.08 (0.78 to 1.50) | 0.96 (0.62 to 1.47) | |
| | | | Tenant at all points | 1.99 (1.10 to 3.59) | 1.13 (0.71 to 1.81) | 1.17 (0.64 to 2.11) | |
| | | 2 | Tenant at one point | 0.83 (0.55 to 1.23) | 0.85 (0.65 to 1.11) | 0.86 (0.61 to 1.21) | 0.72 |
| | | | Tenant at two points | 0.90 (0.54 to 1.49) | 0.99 (0.71 to 1.39) | 0.92 (0.59 to 1.43) | |
| | | | Tenant at all points | 1.48 (0.78 to 2.78) | 0.99 (0.61 to 1.60) | 1.12 (0.60 to 2.08) | |
| Financial hardship accumulation (ref=no or minimal hardship) | 1795 | 1 | Hardship at one point | 2.26 (1.33 to 3.83) | 1.36 (0.89 to 2.08) | 1.57 (0.93 to 2.65) | <0.01 |
| | | | Hardship at both points | 5.18 (1.70 to 15.77) | 1.74 (0.58 to 5.20) | 1.13 (0.23 to 5.66) | |
| | | 2 | Hardship at one point | 1.93 (1.11 to 3.35) | 1.34 (0.86 to 2.08) | 1.59 (0.93 to 2.73) | 0.09 |
| | | | Hardship at both points | 3.90 (1.20 to 12.64) | 1.60 (0.52 to 4.92) | 1.15 (0.23 to 5.84) | |

Model 1: sex-adjusted and Model 2: adjusted for sex, BMI, smoking, alcohol, leisure-time physical activity, GHQ-28 caseness and marital status.
CRP, chronic regional pain; CWP, chronic widespread pain.

to resources (such as better remunerated work) associated with a lower risk of CWP.

Although we found evidence that earlier adulthood may be a sensitive period, it is also clear that the choice of indicator is important as there was limited evidence of an association between occupational class and CWP or CRP. This is surprising given previous findings from the 1958 British birth cohort.[18 19] These previous analyses showed that lower parental occupational class in childhood and own occupational class at age 42 were associated with an increased risk of CWP at age 45 and that the association between own occupational class at 42 and CWP persisted after adjustment for parental occupational class in childhood (and in the case of Macfarlane et al,[18] adjustment for potential confounders/mediators). Reasons for the discrepancy with our findings may be attributable to differences in study design, particularly the ages at which pain was measured. Additionally, due to social, political and cultural changes over the course of the 20th century, conditions at the time of birth in each of the cohorts were different. For example, participants in the NSHD were born in 1946, which was the period immediately after World War II when food rationing was still in effect and important changes in housing regulations had not yet been passed, and so will have experienced poorer absolute economic and environmental conditions in early life than the cohort born in 1958. Further, the effects of occupation may act only contemporaneously in the aetiology of CWP, through the immediate psychosocial or physical demands of the work, for example.[16 17 40 41]

Goosby[20] found that childhood conditions (ie, self-reported parental education, receipt of financial assistance and having gone hungry) were associated with chronic pain prevalence in adulthood (ages 25–64). Differences between this study's findings and our own could be attributable to differences in the indicators of childhood socioeconomic circumstances used. In addition, the associations for childhood SEP indicators observed by Goosby[20] were somewhat attenuated in models that included adulthood SEP (income, education and benefit receipt since age 18). The latter were all associated with chronic pain in the expected directions. The fact that receipt of benefits was associated with chronic pain in Goosby's study[20] is consistent with our finding that financial hardship was associated with CWP.

This study has a number of strengths. The NSHD is largely representative of the population of older adults in Great Britain despite some losses to follow-up,[24 26] meaning that results are likely to be generalisable to this generation. We examined a wide range of SEP indicators with data being collected over a 60-year-period. The study is subject to very limited (if any) recall bias as data were collected prospectively. There is also, therefore, limited risk of exposure or outcome misclassification (ie, participants being incorrectly labelled as lower or higher SEP).

Some limitations are also noted. It was not possible to examine incident pain as no data on CWP or CRP had been ascertained before age 68. As the onset of CWP and

CRP may occur earlier in life and persist into later adulthood, future work will need to establish the temporality of associations. Although we adjusted for a range of covariates, we only did so at one time point. Changes in these variables may be associated with changes in pain state over time and failing to account for their time-varying nature might have resulted in residual confounding. There might also be survivor bias as lower SEP in childhood was associated with lower survival rates into adulthood in NSHD.[42] If so, this would bias associations towards the null and could partially explain why there was no association between childhood SEP and CWP or CRP at age 68. Finally, results might not be generalizable to younger, more ethnically diverse cohorts so the study should be replicated with subsequent generations, particularly the 1958 British birth cohort as participants in that study age, which also contains data on SEP and pain.

These results have important implications for public policy. Given its high prevalence, cost and associated disability, prevention of CWP and CRP should be considered a key public health priority. Policymakers should ensure that socioeconomic adversity across life is addressed in public health interventions, particularly the role that housing, education and financial hardship may play.

**Acknowledgements** The authors are grateful to NSHD study members for their continuing participation in the study. We also thank members of the NSHD scientific and data collection teams.

**Contributors** MAJ had full access to the data and takes full responsibility for the integrity and accuracy of the analysis. MAJ, SGM and RB conceived the idea for this study. MAJ, SGM, RB and RC contributed to study design. MAJ analysed the data and drafted the manuscript. All authors contributed critically to the manuscript and provided final approval of the version to be published.

**Funding** This work was supported by the UK MRC that provides core funding for the MRC NSHD and supported SGM and RC (programme code: MC_UU_12019/4). SGM was also supported by MRC grant MR/L010399/1. MAJ was supported while carrying out this work by a grant from Great Ormond Street Hospital for Children NHS Foundation Trust. RB was supported in part by grant MR/R016372/1 for the King's College London MRC Skills Development Fellowship programme funded by the UK Medical Research Council and grant IS-BRC-1215-20018 for the National Institute for Health Research (NIHR) Biomedical Research Centre at South London and Maudsley NHS Foundation Trust and King's College London.

**Disclaimer** The views expressed are those of the authors and not necessarily those of the MRC, NHS, NIHR or the Department of Health and Social Care. The funders of the study had no role in the study design, data collection, data analysis, data interpretation, writing of the report or decision to submit the article for publication.

**Competing interests** None declared.

**Patient consent for publication** Not required.

**Ethics approval** Ethical approval was received for each data collection; in 2014–2015, this was obtained from the Queen Square Ethics Committee (14/L0/1073) and the Scotland A Research Ethics Committee (14/SS/1009) and written informed consent was obtained from study members.

**Provenance and peer review** Not commissioned; externally peer reviewed.

**Data sharing statement** Data used in this publication are available to bona fide researchers upon request to the NSHD Data Sharing Committee via a standard application procedure. Further details can be found at http://www.nshd.mrc.ac.uk/data and doi: 10.5522/NSHD/Q101; 10.5522/NSHD/Q102 and 10.5522/NSHD/Q103.

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
