## [Reviewer comments · BMJ Open]

ARTICLE DETAILS

TITLE (PROVISIONAL)	Lifetime socioeconomic circumstances and chronic pain in later adulthood: findings from a British birth cohort study
AUTHORS	Jay, Matthew; Bendayan, Rebecca; Cooper, Rachel; Muthuri, Stella

VERSION 1 – REVIEW

REVIEWER	Pamela Andrews Glasgow Caledonian University Scotland, United Kingdom
REVIEW RETURNED	29-May-2018

GENERAL COMMENTS	The authors address an interesting topic, providing evidence of SEP and its impact on developing CWP/CRP in later life. The results of this study will no doubt strengthen the existing literature and will hopefully spark future research addressing some of the limitations highlighted within the text. The study would benefit from a few minor revisions which are listed below. Abstract 1. Design: I think the name of the cohort study used should be included here.2. Primary outcome measures: CRP might be useful to have a brief definition here (ie. >3months in duration but not CWP).3. Results: the sentence “the prevalence of CWP was 13.3% in women and 7.7% in men and that of CRP was 32.3% and 28.7%, respectively” could possibly read better with “the prevalence of CWP was 13.3% in women and 7.7% in men and that of CRP was 32.3% and 28.7%, in women and men respectively”. Introduction 1. CWP prevalence figure: in the CP figure you highlight that it is a UK prevalence estimate. It then reads that the CWP is also a UK estimate, despite the Mansfield paper being a global estimate. In addition, there is also a newer publication with a global CWP prevalence estimate. https://onlinelibrary.wiley.com/doi/abs/10.1002/ejp.10902. The sentence beginning “Understanding risk factors” I think it would strengthen the link by introducing what you mean re risk factors? It reads like there is almost a sentence missing.3. The biopsychosocial sentence, I feel doesn't fit, but possibly with the change in the last point would make this read better. You also don't mention the biopsychosocial model throughout the paper, so I would question of it is needed. Methods
---

	1. Sample: you are missing the abbreviation of NSHD which is used in the PPI section below. Results 1. Is there any reason why you only included the women/men in the results? Would be interested in knowing the overall figures. 2. You also switch between using male/female and men/women, think you should pick just one. References 1. There are a couple of instances that you put the reference at the beginning of a sentence to introduce a paper, I think this should have the authors spelt out with the number at the end. For example: page 5 line 41 "one of these 16 found lower occupational class" I think it should read "MacFarlane et al 2009 found lower occupational class" or MacFarlane and colleagues this may be the referencing style of the journal, but I thought it was worth pointing out. I found the paper interesting and the results and discussions were presented well, I feel that with the above minor amendments it will be a much stronger paper.
--	---

REVIEWER	Jose A. P. da Silva Faculty of Medicine - University of Coimbra – Portugal
REVIEW RETURNED	02-Jun-2018

GENERAL COMMENTS	Excellent and important work presented in a very clear and balanced way. Thank you. A few minor issues:  - Page 7, Line 11. Please present here the abbreviation "NSHD" - Page 12, Line 28. I believe that instead of "Table 4 (and Supplementary Table 5) shows the regression models for each indicator of SEP" it should state "Table 4 (and Supplementary Table 5) shows the RRR obtained through regression models for each indicator of SEP" - Page 13, Line 37. Table 6 is not provided
---

REVIEWER	Eline Coppens KU Leuven, Faculty of Psychology and Educational Sciences, Belgium Department of Psychiatry, The Leuven Centre for Algology & Pain Management, University Hospitals Leuven, University of Leuven, Belgium
REVIEW RETURNED	09-Jul-2018

GENERAL COMMENTS	Introduction Please define CWP and CRP from the start (first paragraph). Second paragraph: ref. 16 found an association between lower occupational class and CRP. In the sentence before you explain these studies investigated associations between SEP and CWP not CRP. Please adjust. Last sentence: why do you make this hypothesis. Please explain. Discussion
--

	P.15 first paragraph: can you try to explain this finding. P.15 second paragraph line 5: I assume you mean 'own occupational class' at age 42? P.15 second paragraph line 10: 'social conditions at birth', please describe P.15 last paragraph: there is clearly a difference in what Goosby measures 'childhood conditions' and what you measured. Goosby measured financial assistance and having gone hungry. Could this explain different findings? P 16 line 3 indicators
--	--

VERSION 1 – AUTHOR RESPONSE

Review 1

The authors address an interesting topic, providing evidence of SEP and its impact on developing CWP/CRP in later life. The results of this study will no doubt strengthen the existing literature and will hopefully spark future research addressing some of the limitations highlighted within the text. The study would benefit from a few minor revisions which are listed below.

Response: We thank the reviewer for their positive assessment of our study.

Abstract

Design: I think the name of the cohort study used should be included here.

Response: The name of the cohort (MRC National Survey of Health and Development) has been moved from the 'Participants' to 'Design' section of the abstract (page 3).

Primary outcome measures: CRP might be useful to have a brief definition here (ie. >3months in duration but not CWP).

Response: A brief definition of CRP is now included in the abstract (page 3).

Results: the sentence "the prevalence of CWP was 13.3% in women and 7.7% in men and that of CRP was 32.3% and 28.7%, respectively" could possibly read better with "the prevalence of CWP was 13.3% in women and 7.7% in men and that of CRP was 32.3% and 28.7%, in women and men respectively".

Response: This sentence in the results section of the abstract has been amended as suggested (page 3).

Introduction

CWP prevalence figure: in the CP figure you highlight that it is a UK prevalence estimate. It then reads that the CWP is also a UK estimate, despite the Mansfield paper being a global estimate. In addition, there is also a newer publication with a global CWP prevalence estimate. <https://onlinelibrary.wiley.com/doi/abs/10.1002/ejp.1090>

Response: Thank you for highlighting this. We have amended the first paragraph of the introduction (page 5) to highlight the relevant figures from the UK systematic review (as our study participants were drawn exclusively from Great Britain) and we have also included reference to the newer international publication (reference 6).

The sentence beginning “Understanding risk factors” I think it would strengthen the link by introducing what you mean re risk factors? It reads like there is almost a sentence missing. The biopsychosocial sentence, I feel doesn’t fit, but possibly with the change in the last point would make this read better. You also don’t mention the biopsychosocial model throughout the paper, so I would question of it is needed.

Response: We have rewritten the end of the first paragraph (page 5) to focus on the risk factors that our study addresses, i.e. SEP. We feel reference to the biopsychosocial model of pain is important, especially for a general audience who may not be familiar with it, as it provides a coherent theory as to why various facets of SEP would be linked with pain. We have amended the paragraph with the aim of clarifying this.

Methods

Sample: you are missing the abbreviation of NSHD which is used in the PPI section below.

Response: We apologise for this omission. We now include the study abbreviation, NSHD, in parentheses when the name of the study is first introduced in the methods section (page 7).

Results

Is there any reason why you only included the women/men in the results? Would be interested in knowing the overall figures.

Response: We thank the reviewer for this suggestion. We gave careful consideration to this but decided against adding the overall figures as these are not meaningful given the important gender differences in prevalence whereby it is more appropriate to present gender specific prevalence.

You also switch between using male/female and men/women, think you should pick just one.

Response: We have changed reference to males/females to men/women so that we now use the term men/women consistently throughout the manuscript. These changes have been made in the third paragraph of the results section (page 12) and in Tables (2 and 4) and Supplementary Tables (1, 2, 3 and 5).

References

There are a couple of instances that you put the reference at the beginning of a sentence to introduce a paper, I think this should have the authors spelt out with the number at the end. For example: page 5 line 41 “one of these 16 found lower occupational class” I think it should read “MacFarlane et al 2009 found lower occupational class” or MacFarlane and colleagues this may be the referencing style of the journal, but I thought it was worth pointing out.

Response: We have made the suggested changes to the text in the second paragraph of the introduction (pages 5-6).

I found the paper interesting and the results and discussions were presented well, I feel that with the above minor amendments it will be a much stronger paper.

Response: We thank the reviewer again for this positive assessment.

Review 2

Excellent and important work presented in a very clear and balanced way. Thank you.

Response: We thank the reviewer for his positive assessment of our work and for acknowledgment of the study's importance.

Page 7, Line 11. Please present here the abbreviation "NSHD"

Response: This abbreviation is now included (page 7).

Page 12, Line 28. I believe that instead of "Table 4 (and Supplementary Table 5) shows the regression models for each indicator of SEP" it should state "Table 4 (and Supplementary Table 5) shows the RRR obtained through regression models for each indicator of SEP"

Response: We have re-written this sentence as suggested (page 12).

Page 13, Line 37. Table 6 is not provided

Response: Thank you for spotting this typographical error. The reference to Table 6 should in fact have been a reference to Table 5. We have checked the other Table numbers and also amended the reference to Table 5 (on page 13, line 15 of the original manuscript) to Table 4. All tables are now correctly cited.

Review 3

I think this is a very solid paper with only a few minor remarks.

Response: We thank the reviewer for this assessment.

Introduction

Please define CWP and CRP from the start (first paragraph).

Response: The definitions of CRP and CWP have been added to the first paragraph of the introduction as requested (page 5).

Second paragraph: ref. 16 found an association between lower occupational class and CRP. In the sentence before you explain these studies investigated associations between SEP and CWP not CRP. Please adjust.

Response: Thank you. We have altered the sentences to make it clear that Macfarlane et al (2009) (now ref 18) also investigated CRP. Power et al (2007) (now ref 19), however, only investigated CWP. They both used the 1958 British Birth Cohort, hence why we mention them together. We hope that by revisiting this paragraph (page 5) we have now clarified this.

Last sentence: why do you make this hypothesis. Please explain.

Response: We hypothesised that there would be an association between lower SEP and higher risk of CWP, but not CRP, on the basis of the findings of Macfarlane et al (2009). In this previous study associations of occupational class with CRP were weaker than those with CWP. An explanation has been added to the manuscript (page 6).

Discussion

P. 15 first paragraph: can you try to explain this finding.

Response: This was a surprising result and we can only speculate as to its explanation. A possible explanation based on the consequences of not having a degree (and how these might differ between men and women) has been added (page 16).

P. 15 second paragraph line 5: I assume you mean 'own occupational class' at age 42?

Response: Thank you for spotting this omission. We did mean 'own occupational class' at age 42 and so have added this information to the relevant sentence on page 16.

P. 15 second paragraph line 10: 'social conditions at birth', please describe

Response: Due to social, cultural and political changes over the course of the twentieth century there were a number of differences in social conditions at birth when comparing cohorts born in 1946 and 1958. We have provided an example of just one of these differences (i.e. the fact that the 1946 cohort were born in the immediate post-war era and so experienced rationing) as an illustrative example of this point, page 17.

P.15 last paragraph: there is clearly a difference in what Goosby measures 'childhood conditions' and what you measured. Goosby measured financial assistance and having gone hungry. Could this explain different findings?

Response: This is a good suggestion and could explain the differences in our findings. We have amended the relevant paragraph on page 17 to acknowledge this.

P 16 line 3 indictors

Response: This has been corrected, page 17.

VERSION 2 – REVIEW

REVIEWER	Pamela Andrews Glasgow Caledonian University United Kingdom
REVIEW RETURNED	15-Oct-2018
GENERAL COMMENTS	The authors addressed all of my comments fully, I have no further comments. I found this a very interesting paper and would recommend for publication.